# Experimental Investigation and Constitutive Modeling of the Uncured Rubber Compound Based on the DMA Strain Scanning Method

**DOI:** 10.3390/polym12112700

**Published:** 2020-11-16

**Authors:** Yong Li, Xunhua Sun, Shoudong Zhang, Yanan Miao, Shanling Han

**Affiliations:** 1College of Mechanical and Electronic Engineering, Shandong University of Science and Technology, Qingdao 266590, China; hexidian@sdust.edu.cn (Y.L.); sunxunhua2020@163.com (X.S.); zhangsd1996@163.com (S.Z.); 2College of Safety and Environmental Engineering, Shandong University of Science and Technology, Qingdao 266590, China; myn2279@163.com

**Keywords:** uncured rubber compound, dynamic thermomechanical analysis tensile strain scanning, modified Burgers model, constant strain rate, viscoelastic

## Abstract

Existing research tends to focus on the performance of cured rubber. This is due to a lack of suitable testing methods for the mechanical properties of uncured rubber, in particular, tensile properties. Without crosslinking by sulfur, the tensile strength of uncured rubber compounds is too low to be accurately tested by general tensile testing machines. Firstly, a new tensile stress testing method for uncured rubber was established by using dynamic thermomechanical analysis (DMA) tensile strain scanning. The strain amplitude was increased under a set frequency and constant temperature. The corresponding dynamic force needed to maintain the amplitude was then measured to obtain the dynamic force-amplitude curve observed at this temperature and frequency. Secondly, the Burgers model is usually difficult to calculate and analyze in differential form, so it was reduced to its arithmetic form under creep conditions and material relaxation. Tensile deformation at a constant strain rate was proposed, so the Burgers model could be modified to a more concise form without any strain terms, making mathematical processing and simulating much more convenient. Thirdly, the rate of the modified Burgers model under constant strain was in good agreement with the test data, demonstrating that the elastic stiffness was 1–2 orders of magnitude less than the tensile viscosity. In the end, it was concluded that large data dispersion caused by the universal tensile test can be overcome by choosing this model, and it may become an effective way to study the tensile modeling of uncured rubber compound.

## 1. Introduction

The rubber-fiber composite has become the most preferred type of skeleton in the rubber industry. This is due to its excellent mechanical properties, as can be seen in common rubber products such as tires, air springs, and transport belts [1,2,3,4]. Researchers can design different rubber-fiber composite products according to their wishes, but the actual performance of these composites can be far from desired. According to the U.S. Department of Transportation, there have been 1718 tire recalls in recent years, and problems directly related to rubber-fiber composite materials, including fiber delamination and interlaminar defects, account for 49.6% of such recalls (Figure 1). The main reason is because the existing research focuses on macro composite mechanics. Researchers have focused on the composite structure adopting equivalent stress and ignored the interface problem [5]. The rubber-fiber composite’s fine internal cord structure, combined with rubber, should be further investigated. The uncured rubber compound has not been cross-linked, so its tensile strength is low. The rubber-fiber interface has not formed a Cu-S chemical bond and is only bound by the van der Waals force, so the bonding strength is low too, which makes it difficult to detect by conventional means. The measurement and research of uncured rubber can deepen the understanding of its mechanical behavior. Whether the mold cavity can be filled in the rubber molding process and the flow behavior of uncured rubber during the molding process will greatly affect the quality of the finished rubber composite. However, there is still a lack of research in this area. Therefore, it is urgent to find a reliable measurement method for uncured rubber. At present, there is no direct testing method for uncured rubber compounds with excellent reproducibility. In the building process, there exists highly complex nonlinear deformation. Thus, indirect means make it difficult to reflect the actual stress-strain state and cannot provide reliable data support for establishing the model. It is difficult to obtain accurate stress-strain values using common measuring methods to measure uncured rubber because of the large data dispersion caused by problems with the clamping method and accuracy. 

The difficulty in the constitutive study of uncured rubber compounds lies in the testing of its tensile properties. Compared with the mature constitutive model of vulcanized rubber, there are few mechanical studies on uncured rubber compounds [6,7]. Kaliske et al. [8,9] tested the tensile and shear properties of uncured rubber compounds, and proposed an internal variable viscoplastic constitutive model based on microspheres. Li et al. [10,11] adopted the non-contact automated grid method to conduct a tensile test on the uncured rubber sample to obtain its full-field large deformation, and proposed a three-parallel network model consisting of a hyperelastic part and two elastic-viscoplastic parts. Feng et al. [12,13] designed three different mechanical properties experiments on uncured rubber compounds and proposed the Generalized Maxwell viscoelastic model. Uncured rubber compounds have low strength and poor shape precision. The key to establishing a model for uncured rubber compounds also lies in the initial sample retention, testing method, and measurement accuracy.

Unlike Newtonian fluid, uncured rubber compounds show both viscous flow and elastic deformation during processing. Its viscosity and modulus are strongly dependent on the action rate of external forces. Shear viscosity is closely linked to the processing property and mixing uniformity of rubber. In previous literature, the research on shear viscosity is clearer and more in-depth [14,15,16]. Li et al. [17] proposed a four-parameter rheological model based on the existence of yield stress and shear-thinning behavior of the uncured rubber compound. The tensile viscosity reflects the “elasticity” of the material, and the tensile viscosity is often 102–103 times that of the shear viscosity, which is mainly brought about by the first normal stress difference [18]. Lin et al. [19] measured the SBR melt in the uniaxial tensile process, indicating that the tensile viscosity showed great elasticity. Malkin et al. [20] noted that polymer melts are similar to rubber state when stretched. Qu et al. [21,22,23] used volume pulsating tensile flow field to blend and modify microfiber and granular polymer composites, which improved dispersion and tensile strength, while Xie et al. [24] studied the nucleation and growth of pores in the uniaxial tensile process of polypropylene film at different temperatures. However, there is little research on the tensile constitutive model of uncured rubber compound, which is an important parameter affecting the permeability of the viscoelastic body, and it is worth further study.

## 2. Materials and Methods 

### 2.1. Materials 

The uncured rubber compound studied in this paper was provided by Shandong University of Science and Technology Key Laboratory. The sample ingredients used in this test comprised natural rubber (NR), cis-1,4-polybutadiene rubber (BR), carbon black (CB) N234 and N115, silica, sulfuric agent, and anti-aging agent. Details are provided in Table 1.

To obtain stable and reliable samples, the uncured rubber compound needed to be treated in two steps before the experiment. Firstly, three-stage mixing was performed in a mixer. Then, the sample sheet was pressed through the gap of the rollers, and the sample cuboid shape was cut to be 6.00 mm wide, 1.80 mm thick, and 50 mm long. 

### 2.2. Devices

Dynamic thermomechanical analysis (DMA) measures the relationship between the mechanical properties of viscoelastic materials and time, temperature, as well as frequency [25]. The sample was deformed under the action and control of a periodically changing sinusoidal mechanical stress [26]. By measuring the corresponding deformation amplitude and hysteresis of the material, relevant characteristic curves such as energy storage modulus, loss modulus, and loss factor could be calculated and obtained. This could characterize the change of viscoelastic energy of the material with temperature, frequency, as well as relevant characteristic temperature points such as Tg and Tm [27]. DMA has high precision and superior repeatability. For example, the DMA 242 E used in the experiment had broad technical parameters: the temperature range was from −170 to 600 °C, the heating rate was from 0.01 to 20 K/min, the frequency range was from 0.01 to 100 Hz, the controllable strain accuracy range was from −240 to +240 μm, and the modulus range was from 10^−3^ to 10^6^ MPa, which ensured more accuracy than provided by the universal tensile testing machine.

The traditional measurement of uncured rubber is to measure its viscosity with a Mooney viscometer, which cannot express its viscoelasticity well. It is difficult to obtain an accurate value of stress and strain in the usual uncured rubber tensile measurement method, due to the intractable problem of large data dispersion caused by the clamping method and measurement inaccuracy. On the other hand, DMA has the following advantages, among others: high accuracy; excellent repeatability; the ability to analyze solids, pastes, and liquids; has extensive sample holders; the ability to provide auxiliary sample sizes; and the ability to direct output of stress in measurement. However, DMA is rarely used to measure the properties of uncured rubber to model and verify it, so a new tensile stress testing method for uncured rubber was established by using DMA dynamic tensile strain scanning. Its strain amplitude increased during a specific step from 0 to 5% under a set frequency and constant temperature, and the corresponding dynamic force needed to maintain the strain amplitude was measured to obtain the corresponding dynamic force-strain curve (Figure 2). By using this model, large data dispersion caused by clamping and the imprecise measurements of the universal tensile test machine can be overcome. This model is thus an effective way to study uncured rubber compounds.

The dynamic strain scanning method has the following advantages: (1) As is well known, the lower the temperature, the harder the rubber, so clamping the rubber sample at −30 °C with a contact force of less than 0.5 N can prevent excessive gripping deformation. This will prevent excessive local stress, which results in large testing data dispersion. (2) The sample is a simple cuboid, convenient for cutting and helping to maintain shape accuracy. (3) Due to the measurement range of the sample being less than 30 mm, this method can reduce deformation caused by gravity. (4) When the viscoelastic properties of uncured rubber compounds are measured under dynamic strain, the apparent strength is considerably enhanced and the relative measurement accuracy is accordingly improved by sinusoidal stress. (5) The strain loading step is 0.1%, which is more accurate than that of the general tensile testing machine, improving the reproducibility of data. Because of the above advantages, the DMA strain scanning as a new measurement method of the uncured rubber should be further investigated and standardized.

### 2.3. Methods

In this experiment, a new measuring method of uncured rubber, the DMA strain scanning method, was used to assess the properties of uncured rubber. The specific operation was divided into four steps.
In order to maintain the shape stability, clamping was done at −30 °C, and 0.4 N contact force and 7% pre-strain was applied (Figure 2).The strain loading method started from 0.1% strain at a temperature of −20 °C with a frequency of 10 Hz.According to the stride length of 0.1%, dynamic forces needed to maintain different strains were applied (Figure 3).According to the monotonic results of the stress-strain behavior of the uncured rubber compound, the dynamic force and strain test equations at a fixed temperature and frequency were obtained.

DMA has the advantages of high accuracy, excellent repeatability, analysis of solid, paste, and liquid, extensive sample holders, auxiliary sample size, direct output of stress in measurement, etc. Theoretically, the dynamic stress and strain changes of the uncured rubber compound during the tensile test can be measured more accurately, which is very important to understand the properties of uncured rubber, and we needed to verify it in this experiment.

Monotonic results of the stress-strain behavior of the uncured rubber compound are shown in Figure 4. The empirical equation can be obtained by cubic fitting of the obtained stress-strain curve, which is more accurate than the quadratic fitting. The model is in good agreement with the actual test data and can be directly used as the dynamic force and strain testing equation at fixed temperature and frequency.
(1)$$\sigma =1.547\varepsilon^{3}-21.098\varepsilon^{2}+103.545\varepsilon +0.586$$

## 3. Results and Discussion

In the cutting and building process of the uncured rubber compound, it was kept at a temperature of −20 °C and linear macromolecules had not been cross-linked, so its viscoelasticity was closer to a fluid than a solid. Therefore, the four-element model (Burgers model) was chosen for analysis. The Burgers model is essentially formed by connecting a Maxwell model representing fluid and a Kelvin model representing solid in series (Figure 5). The Burgers model is usually difficult to calculate and analyze in differential form, so it can be reduced to the arithmetic form under the conditions of creep and material relaxation [28,29,30]. The tensile deformation at constant strain rate was proposed, so the Burgers model could be modified in a more concise form without any strain term.

The strain of this model consists of three parts: elastic deformation $\varepsilon_{1}$, viscous flow deformation $\varepsilon_{2}$, and viscoelastic deformation $\varepsilon_{3}$. Their stress relationships are as follows:(2)$$\sigma =E_{1}\varepsilon_{1}=\eta_{2}\varepsilon_{2}=E_{3}\varepsilon_{3}+\eta_{3}\varepsilon_{3}$$
(3)$$\varepsilon =\varepsilon_{1}+\varepsilon_{2}+\varepsilon_{3}$$

Laplace transform of Equations (2) and (3) is applied as follows:(4)$$\sigma +\left(\frac{\eta_{2}}{E_{1}}+\frac{\eta_{2}+\eta_{3}}{E_{2}}\right)\dot{\sigma }+\frac{\eta_{2}\eta_{3}}{E_{1}E_{2}}\ddot{\sigma }=\eta_{2}\dot{\varepsilon }+\frac{\eta_{2}\eta_{3}}{E_{2}}\ddot{\varepsilon }$$

Under constant tensile strain, $\dot{\varepsilon }=C$ is constant and $\ddot{\varepsilon }=0$. Equation (4) can be reduced to
(5)$$\sigma =-\left(\frac{\eta_{2}}{E_{1}}+\frac{\eta_{2}+\eta_{3}}{E_{2}}\right)\dot{\sigma }-\frac{\eta_{2}\eta_{3}}{E_{1}E_{2}}\ddot{\sigma }+\eta_{2}C$$

Equation (5) is the Burgers equation at constant strain rate, which has the prominent feature of having no strain term. This is convenient for mathematical processing and simulating.

### 3.1. Determination of Zero Point of Action Time

During the test, since the experiment started from the clamping of the sample and the pre-loading stage, it was necessary to determine the initial time of the stretching experiment, namely, the time zero point. According to the Burgers model, at the moment of force application, only spring 1 will be deformed due to the action of sticky pots. According to Hooke’s law, the elastic constant $E_{1}$ can be obtained by using the numerical value of this moment. Therefore, it is very important to determine the time zero point when the force applied to the sample begins. However, there may be a time lag due to the timing of the experimental instrument at the beginning of the experiment. Therefore, the first timing point in the experimental data was set as the corresponding point at the beginning of stretching for 1 s to improve the accuracy. The software Origin was used to take the measured time as the *Y*-axis, and the time axis starting from 1 was the *X*-axis for fitting (Figure 6). Thus, the required time zero point *t*_0_ was obtained ($t_{0}=512.14\;s$).

### 3.2. Determination of Elastic Constants $E_{1}$

At the moment of applying the tensile force, only spring *E*_1_ produced the instant deformation due to the action of dashpot $\eta_{2}$ and $\eta_{3}$. The elastic constants *E*_1_ can be calculated from the stress-strain relationship at time *t*_0_.
(6)$$E_{1}=\frac{d\sigma }{d\varepsilon }|{}_{t_{0}^{+}}=\frac{d\sigma /dt}{d\varepsilon /dt}|{}_{t_{0}^{+}}=\frac{\dot{\sigma }}{\dot{\varepsilon }}|{}_{t_{0}^{+}}$$

In order to obtain stress rate $\dot{\sigma }$ and strain rate $\dot{\varepsilon }$, the following formula was adopted:(7)$$f^{\prime }\left(x\right)=\frac{-f\left(x_{i+2}\right)+8f\left(x_{i+1}\right)-8f\left(x_{i-1}\right)+f\left(x_{i-2}\right)}{12h}$$

Because the empirical equation of the stress-strain data was obtained by cubic fitting, the stress rate and strain rate needed to be fitted quadratically. Using Origin and taking the first 10 data of the stress rate obtained from Equation (7) at the beginning period of ε≤1%, the quadratic fitting was carried out and extended to the time zero, and the following formula was obtained (Figure 7).
(8)$$\dot{\sigma }=2.72\times 10^{-6}t^{2}-4.35\times 10^{-3}t+2.14$$

Bringing in $t_{0}=512.14\;s$, the following could be obtained: $\dot{\sigma }=0.62\;\mathrm{KPa}/s$. In the same way, the strain rate $\dot{\varepsilon }$ could be obtained as follows: (9)$$\dot{\varepsilon }=-8.32\times 10^{-8}t^{2}+1.10\times 10^{-4}t-0.029$$

Bringing in $t_{0}=512.14\;s$, the following could be obtained: $\dot{\varepsilon }=0.0052\;s^{-1}$, thereby obtaining $E_{1}=119.29\;\mathrm{KPa}$.

### 3.3. Determination of Other Parameters

Because this experiment adopted a uniform speed tension method of measurement, the slope of the strain is strain rate by linear fitting, so $C=\dot{\varepsilon }=0.0066$ was obtained (Figure 8). At the same time, according to Equation (7), $\ddot{\sigma }$ was obtained. Polynomial fitting of Equation (4) was obtained by Origin as follows:(10)$$\sigma =-263.81\dot{\sigma }-61.42\ddot{\sigma }+195.39$$

Dynamic stress-strain data can be obtained by the DMA strain scan, stress rate and second derivative of stress can be obtained by differentiating with respect to time, and Formula (10) can be obtained by linear regression fitting with Origin according to Formula (5) after obtaining the data.
(11)$$\eta_{2}C=195.39$$
(12)$$\frac{\eta_{2}}{E_{1}}+\frac{\eta_{2}+\eta_{3}}{E_{2}}=263.83$$
(13)$$\frac{\eta_{2}\eta_{3}}{E_{1}E_{2}}=61.42$$

By substituting $E_{1}=119.29\;\mathrm{KPa}$ and C=0.0066 into the above equation, we obtained $E_{2}=1739.73\;\mathrm{KPa}$, $\eta_{2}=29425.75\;\mathrm{KPa}\cdot s$, and $\eta_{3}=433.19\;\mathrm{KPa}\cdot s$. Therefore, we came to the conclusion that the elastic stiffness was 1–2 orders of magnitude less than the tensile viscosity. This indicated that the uncured rubber compound was closer to a fluid than a solid, which is consistent with the assumption of the Burgers model.

### 3.4. Simulink Simulation

According to the parameters and initial conditions determined by Equation (4) in Section 3.1, Section 3.2 and Section 3.3, the Simulink simulation (Figure 9) was put in place to obtain the comparison between the stress-time theoretical curve and the testing data. The initial values were $\dot{\varepsilon }=0.0066$, ${\dot{\sigma }}_{0}=0.624\;\mathrm{KPa}/s$, $\sigma_{0}=0\;KPa$, $t_{0}=512.14\;s$.

Figure 9 shows the Simulink simulation model. Using the Simulink module in MATLAB, the Burgers model formula was modeled. Since the elastic constants and viscosity values in the Burgers model formula had been obtained in the previous part of the experiment, the Burgers model’s stress-time curve could be simulated by substituting the Burgers model into the Simulink model.

Figure 10 shows the comparison of test data and theoretical curves. In the figure, the test data curve is the actual stress time curve measured by DMA, and the theoretical curve is the curve output by Simulink simulating the Burgers model. The consistency of the two curves is good, which proves the feasibility of measuring the performance of uncured rubber by DMA.

## 4. Conclusions

The traditional measurement of uncured rubber is to measure its viscosity with a Mooney viscometer, which cannot express its viscoelasticity well. It is difficult to obtain the accurate value of stress and strain in the common uncured rubber tensile measurement method, due to large data dispersion caused by clamping and measurement inaccuracy. As mentioned above, DMA has the following advantages, among others: high accuracy; excellent repeatability; the ability to analyze solids, pastes, and liquids; possesses extensive sample holders; provides for auxiliary sample size; and can direct the output of stress in measurement. However, DMA is rarely used to measure, model, and verify the properties of uncured rubber, so this paper presents a new method of uncured rubber tensile stress test, that of the DMA strain scanning method.

Without crosslinking by sulfur, the tensile strength of the uncured rubber compound is too low to be accurately tested by the general tensile testing machine. A new tensile stress testing method for uncured rubber, which detects the dynamic force needed to maintain different dynamic strain amplitudes at a specific frequency, was established by using DMA tensile strain scanning. In DMA tensile strain scanning, the strain amplitude was increased under a set frequency and constant temperature. The corresponding dynamic force needed to maintain the amplitude was measured to obtain the dynamic force-amplitude diagram at this temperature and frequency. The outstanding features are high precision and good data consistency. The empirical equation can be obtained by cubic fitting of the obtained stress-strain curve. 

Using tensile strain at a constant rate, the modified Burgers model is in good agreement with the test data. Under this condition, the first-order strain rate $\dot{\varepsilon }$ is constant and the second-order strain rate $\ddot{\varepsilon }$ is zero. The isotropic parameters of the modified Burgers model can be obtained by the polynomial fitting. Compared with the experimental data from the theoretical curve of the modified Burgers model by Simulink, they have good consistency, which shows that the model is feasible.

In conclusion, the elastic constants and viscosity parameters of the Burger model can be obtained by processing the stress-strain parameters obtained by DMA strain scanning with Origin and bringing them into the simulation Burger model in Simulink for simulation. The output theoretical curves agree well with the experimental data curves obtained by the DMA strain scanning method, which proves the feasibility of DMA strain scanning in measuring the properties of uncured rubber and verifies the rationality of the new modified Burger model. Both the experimental data and the theoretical model show that it is feasible to study the tensile properties of uncured rubber compounds using the DMA strain scanning method. The DMA strain scanning method is thus worth further study and standardization.

## Figures and Tables

**Figure 1 polymers-12-02700-f001:**
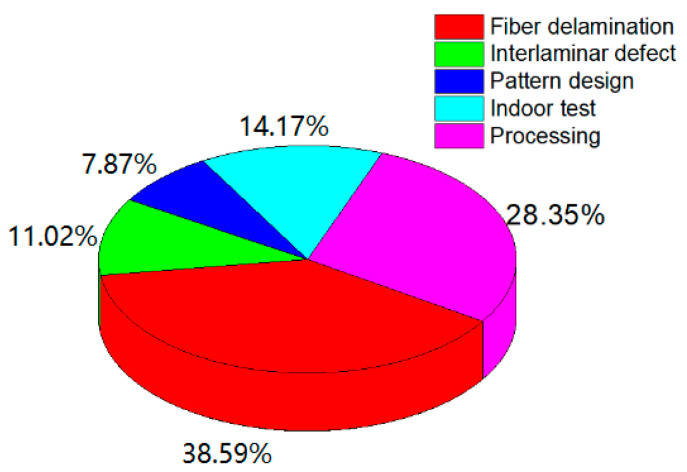
Rubber-fiber composite issues are directly related to approximately half of all tire problems. From arfc.org.

**Figure 2 polymers-12-02700-f002:**
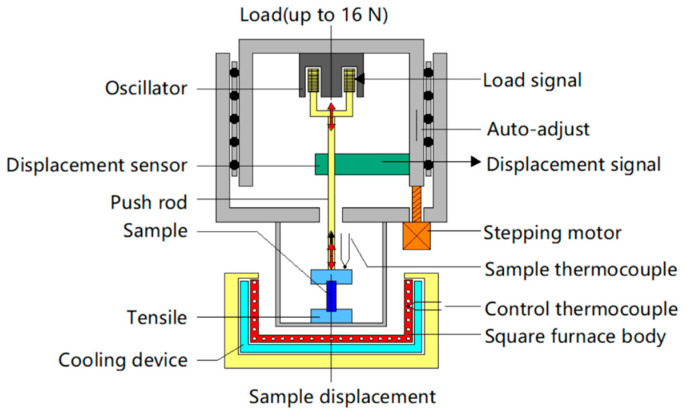
Dynamic thermomechanical analysis (DMA) tensile strain scanning with high precision.

**Figure 3 polymers-12-02700-f003:**
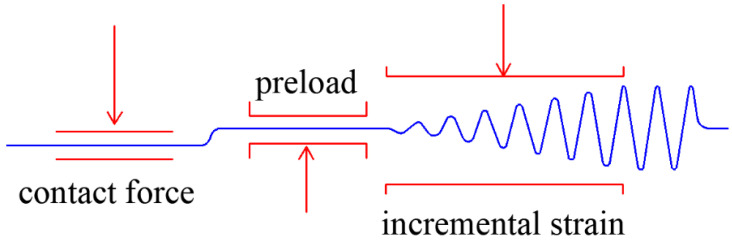
DMA strain scanning force loading process.

**Figure 4 polymers-12-02700-f004:**
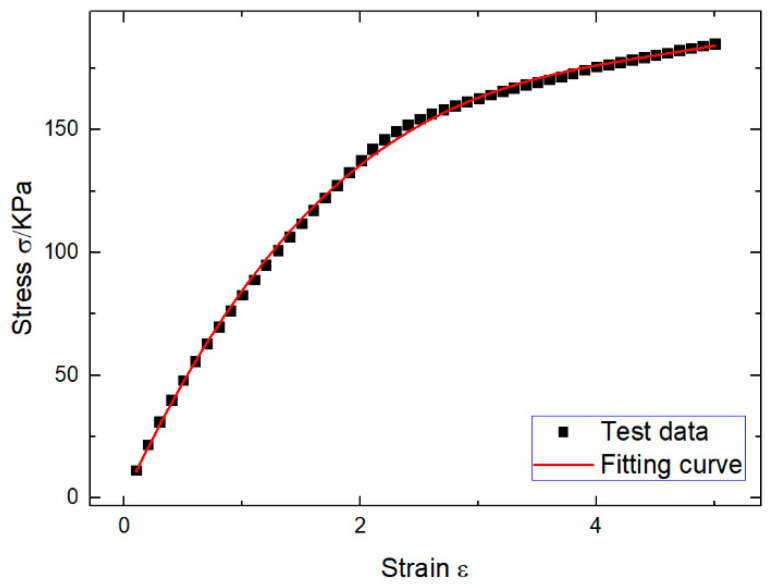
Stress-strain curve of the DMA tensile strain scanning.

**Figure 5 polymers-12-02700-f005:**
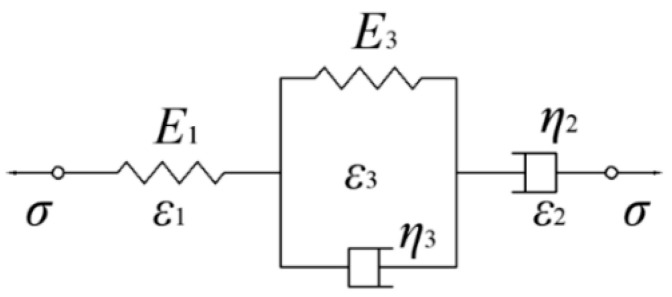
Burgers model.

**Figure 6 polymers-12-02700-f006:**
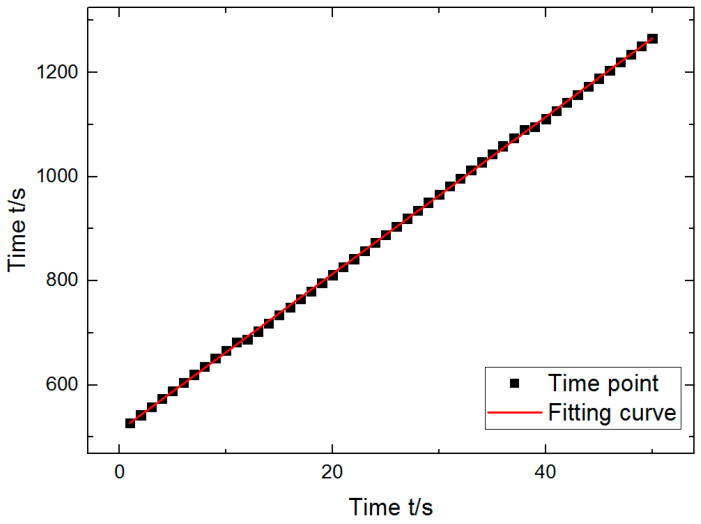
Time zero fitting curve.

**Figure 7 polymers-12-02700-f007:**
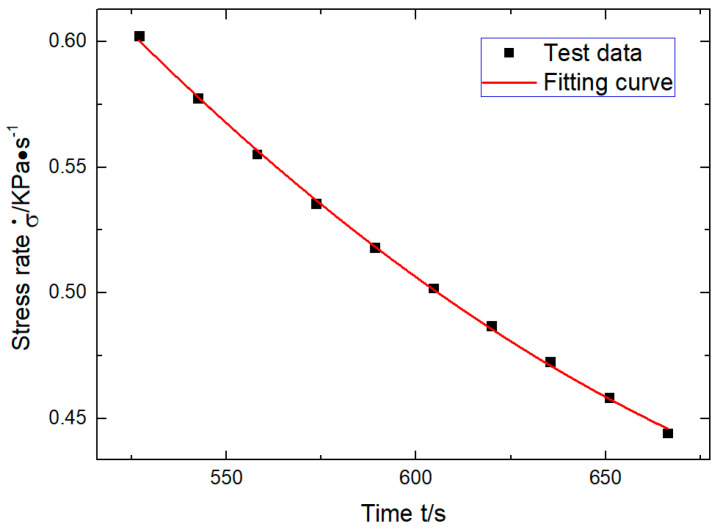
Obtaining zero-time stress rate by quadratic fitting epitaxy of stress rate.

**Figure 8 polymers-12-02700-f008:**
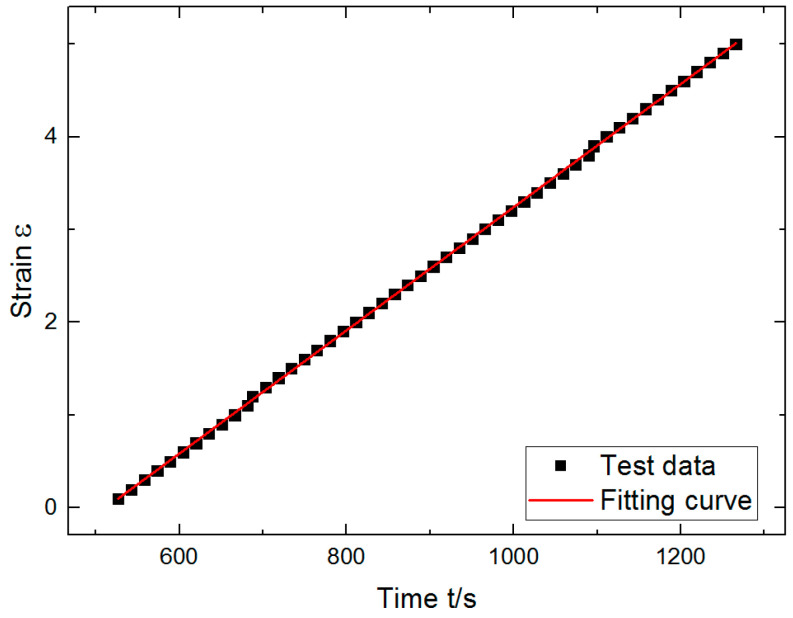
Strain time fitting curve.

**Figure 9 polymers-12-02700-f009:**
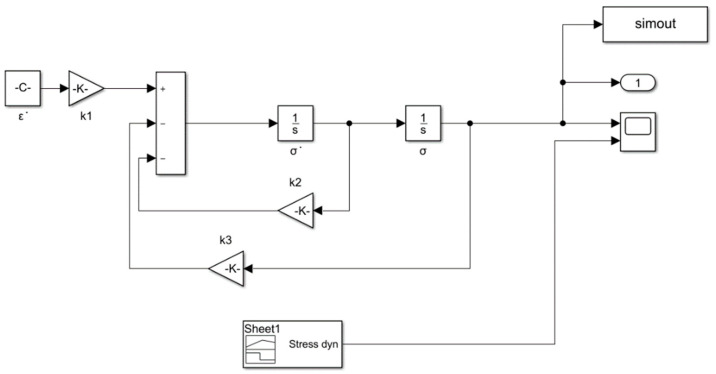
Simulink simulation.

**Figure 10 polymers-12-02700-f010:**
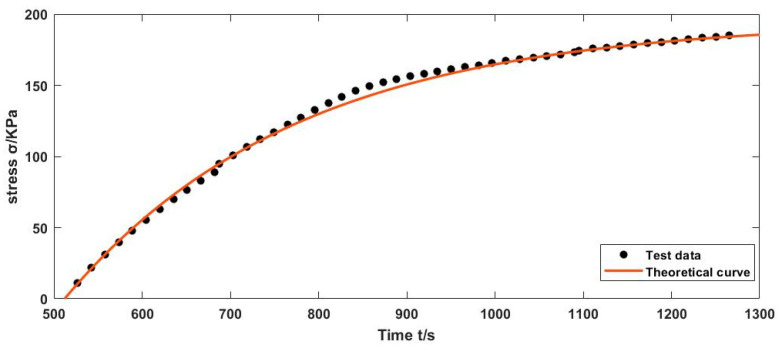
Comparison of the theoretical curve with our test data.

**Table 1 polymers-12-02700-t001:** Amounts of ingredients in the sample.

Ingredient	Amount (phr)
natural rubber (NR)	90
cis-1,4-polybutadiene rubber (BR)	10
carbon black (CB) N234	38
carbon black (CB) N115	10
silica	10
sulfuric agent	4.4
anti-aging agent	2

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
