# Peer review of "Experimental Investigation and Constitutive Modeling of the Uncured Rubber Compound Based on the DMA Strain Scanning Method"

_polymers, 2020, doi:10.3390/polym12112700_

Round 1

Reviewer 1 Report

"Constitutive Modeling of the UncuredRubber Compound Based 3on the DMA Strain Scanning Method"
 The manuscript is interesting but needs some corrections:

1.please describe the elements of the scientific novelty
2.figure 8,9,10 must be accurately described
3.please describe the summary in conclusion

4.please describe chapter "3.1 Determination of zero point of action time better"

Reviewer 2 Report

This paper presents an experimental testing on uncured rubber compound samples by means of the DMA stain scanning method. Then, the experimental results are fitting with the Burgers model and the material parameters of the model are obtained.

I have some serious concerns about the paper.

  1. I understand that equations (8) and (9) are the fitting of the experimental curves stress rate vs time and strain rate vs time, respectively. The last one is shown (Figure 8) but not the first one. I think that the strain rate vs time curve must be represented. Otherwise is impossible to see the linear fitting to obtain t0 and C. With respect to E1, I suppose that the numerical value coincides with the slope at the origin in Figure 6. I also understand that equation 1 is the fitting of the experimental stress-strain curve
  2. The advantages of the DMA should be written before the results. I suggest in line 116, before the description of the results, and perhaps equation 1 and figure 6 should be moved to point 3. Results and Discussion
  3. The units of stress rate in line 178 and strain rate in line 181 are not correct. they are KPa/s and s-1.
  4. I can not understand how equation 10 have been derived. This is very important because from this equation the other material parameters are obtained.
  5. Why it is assumed that the strain rate is constant?
  6. I do not see the necessity of showing figure 3, especially if the testing number are not described
  7. Figure 4 is absolutely unnecessary. But if the authors want to show the shape of the specimen, at least, they should maintain the aspect ratio
  8. I do not see what equation 7 contributes to the paper.
  9. How many tests have been performed? Reading the paper I would say that only one sample has been tested
  10. I can not follow the flux of Figure 9

Reviewer 3 Report

The paper deals with “Experimental Investigation and Constitutive Modeling of the Uncured Rubber Compound Based on the DMA Strain Scanning Method”.

  • This can be an interesting and scientifically relevant to publish in Polymers. The references are up to date. However, one of the most important drawbacks concerns the originality and novelty of the paper. It is difficult to see what the difference is between their work and the literature. Of course, it is mentioned in the introduction part, but it is not clear enough. Therefore, it must be highly desirable to provide more discussion to get insight on the originality of the paper.
  • Please provide explanation of why authors studied “Uncured Rubber Compound” using the DMA, and what will be the benefit to investigate this?
  • Authors can provide references for the Figures if any.
  • “Materials and Methods” part is not clear and can be re-arranged.
  • Figure 2, please provide the description for the “new tensile stress testing method“
  • Authors, please explain Figure 3, and provide the difference between sample 1, 2, 3, and 4.
  • It seems that Figure 4 is not important to present.
  • Figure 6, additional discussion to support this Figure is required.
  • Line 204 to 205, authors stated “From Figure 10, it can conclude that the theoretical curve of the modified Burges model at 205 constant strain rate is in accordance with the experimental data”. More information, discussion for Figure 10 are required. Additional discussion to support this statement would be useful.
  • Finally, please do a thorough editing effort in the English language and make sure that format, spaces and proper grammar rules are followed throughout the manuscript.

Reviewer 4 Report

The authors reported the tensile testing of uncured rubber by DMA method. The authors presented some mathematical models to characterize the tensile strength of the rubbers.

Please consider some comments/ observations which might improve the quality of the manuscript:

  1. Phrase from introduction section “The main reason is due to the research focus on the macro composite mechanics, that is to say, the complex structure becomes the simplified rebar model and the interface layer is idealized [5], so its internal cord fine structure combined with rubber should be further investigated” was not clear enough; The authors are requested to reformulate.
  2. The symbols explaining the system Cu- S was represented as “-Cu=S- chemical bond” as between Cu and S there is a covalent bond; the authors are requested to represent the Cu-S system properly to avoid misunderstandings.
  3. A clear statement of the purpose of the study and its motivation is missing; therefore, the manuscript looks like an enumeration of various tests and models. The authors are requested to provide details on the innovative part of the study.
  4. Materials and methods section provided details of DMA results. However, this part did not provide clear details on the used materials, tests performed, used devices, parameters together with their explanatory notes, calibration curve, etc. The authors are demanded to reformulate this part and to reorganize it accordingly; the authors might take as an example previously published manuscript for guidance.
  5. The results parts contain details on the mathematical models and some results. However, there is no logical scheme of these results to follow and to reproduce the results. The authors are requested to reorganize the results part to be clearly the developments and the obtained outcomes.
  6. The authors claimed that they developed a new tensile testing method; they should highlight more about this method as it was not clear from de description of methods.
  7. The authors presented some tensile strength results; however, the software used for the calculation of these results was not mentioned. The authors are requested to provide details.

Round 2

Reviewer 1 Report

I accept the manuscript in present form

Author Response

Thank you for your careful and  hard work. This article has been modified in accordance with your comments.

Reviewer 2 Report

The questions have been answered and incorporated into the text.

Author Response

(The authors gave the same response as above.)

Reviewer 4 Report

The authors answered the demanded queries so that the manuscript's quality got improved. However, there are still minor typing mistakes and English words their corrections are needed.  

Subtitle from 2.2. was written with small letter; the authors are requested to double check the typing and writing style in order to be uniform throughout the manuscript. 

Author Response

(The authors gave the same response as above.)
